# COVID-19 Diagnostic Strategies Part II: Protein-Based Technologies

**DOI:** 10.3390/bioengineering8050054

**Published:** 2021-04-28

**Authors:** Tina Shaffaf, Ebrahim Ghafar-Zadeh

**Affiliations:** 1Biologically Inspired Sensors and Actuators Laboratory (BioSA), York University, Toronto, ON M3J1P3, Canada; tshaffaf@yorku.ca; 2Department of Biology, Faculty of Science, York University, Toronto, ON M3J1P3, Canada; 3Department of Electrical Engineering and Computer Science, Lassonde School of Engineering, York University, Toronto, ON M3J1P3, Canada

**Keywords:** COVID-19 detection, protein-based tests, SARS-CoV-2, point-of-care (POC) detection, serological tests, antigenic tests, rapid diagnostic tests, LFIA, ELISA, protein microarray

## Abstract

After the initiation of the current outbreak, humans’ lives have been profoundly impacted by COVID-19. During the first months, no rapid and reliable detecting tool was readily available to sufficiently respond to the requirement of massive testing. In this situation, when the development of an effective vaccine requires at least a few months, it is crucial to be prepared by developing and commercializing affordable, accurate, rapid and adaptable biosensors not only to fight Severe Acute Respiratory Syndrome Coronavirus 2 (SARS-CoV-2) but also to be armed to avoid the pandemic in the earliest stages in the future. The COVID-19 diagnostic tools are categorized into two main groups of Nucleic Acid (NA)-based and protein-based tests. To date, nucleic acid-based detection has been announced as the gold-standard strategy for coronavirus detection; however, protein-based tests are promising alternatives for rapid and large-scale screening of susceptible groups. In this review, we discuss the current protein-based biosensing tools, the research advances and the potential protein-detecting strategies for COVID-19 detection. This narrative review aims to highlight the importance of the diagnostic tests, encourage the academic research groups and the companies to eliminate the shortcomings of the current techniques and step forward to mass-producing reliable point-of-care (POC) and point-of-need (PON) adaptable diagnostic tools for large-scale screening in the future outbreaks.

## 1. Introduction

The spread of the Severe Acute Respiratory Syndrome Coronavirus 2 (SARS-CoV-2) has resulted in a life-threatening novel respiratory disease worldwide [1]. By 5 March 2021, the total number of the confirmed coronavirus disease 2019 (COVID-19) cases and the deaths have been reported as 115,289,961 and 2,564,560, respectively [2]. The main concern regarding the prompt spread of this virus is that a considerable portion of the infected cases do not experience the typical known symptoms of fever, fatigue and dry cough in the first stage and only symptomatic cases of COVID-19 are being identified and isolated [3]. Such pre-symptomatic or asymptomatic cases are not informed of the being carriers and accelerate the community spread by their presence in the society. To date, a few companies have developed vaccines to protect people from the infection caused by COVID-19. Although these vaccines have successfully received the U.S. Food and Drug Administration’s (FDA) approval, their long-term effects, side effects, and the effectiveness of them are some other challenges requiring more time and evaluation after the initiation of vaccination in the real world. Considering the serious and fatal effects of COVID-19, reliable rapid, accurate and affordable diagnostic tests are urgently required not only for detection of the infection in the earliest stages but also to monitor the disease and conducting convalescence studies to control this outbreak.

Based on FIND, as of March 5, at least 1130 diagnostic tests have been submitted to various regulatory organizations in different countries, about 1030 tests have received emergency authorization and are manufactured [4]. Two main diagnostic categories are employed for sensing the coronavirus 2019; the first group targets viral nucleic acid and the second one captures specific polypeptides either in the virus structure or in the blood of the suspected people. The present article provides a narrative review to discuss the current protein-based technologies for COVID-19 detection and challenges associated with each technology. In the rest of the paper, Section 2 describes the structure of the coronavirus 2019 and the immune response to SARS-CoV-2 specific antigens, Section 3 explains both laboratory-based and rapid PC SARS-CoV-2 serological tests, Section 4 discusses antigenic tests, and the last section includes the potential protein-based strategies to be developed for COVID-19 detection in the future.

## 2. SARS-COV-2 Virology

The genetic material of the SARS-CoV-2 is an RNA including six open reading frames (ORFs) responsible for the production of the nucleocapsid (N), spike (S), membrane (M) and a small envelope (E) structural protein subunits; the rest of the genes produce 16 functional proteins such as RNA dependent RNA polymerase (RdRP) and helicase [5]. N protein attaches to the coronavirus genetic RNA and forms nucleocapsid. Spike glycoproteins (S1 and S2 subdomains) are surface densely glycosylated proteins specifying the type of the infected host and are involved in various tendencies to different tissues [6]. Angiotensin-converting enzyme 2 (ACE2), expressed in human endothelial cells in the lung, intestine, heart and kidney, is a surface protein and a functional receptor for coronavirus [7]. This protein acts as a direct binding site for virus S protein, the receptor-binding domain (RBD) of the S1 subunit of S protein directly contacts the ACE2 receptor with a high affinity and has crucial roles in attachment and fusion of the virus through the ACE2-containing host cells [8]. M protein has the main role in forming new virus particles and could insert some proteins from the host to the viral envelope [9]. The E protein is the smallest structural protein having roles in coronavirus assembly and pathogenesis [10]. The arrangement of these four proteins is different among coronaviruses, but their presence is critical for infectious characteristics of SARS-CoV-2. After the entry of the coronavirus into the human body, the immune system demonstrates a prompt defensive response to the viral antigens and produces specific antibodies to fight the disease [11]. The coronavirus is capable of inducing a variety of disease statuses from asymptomatic infection to mild and severe respiratory failures in some cases. The main symptoms associated with COVID-19 include fever, dry cough, shortness of breath and fatigue. In the COVID-19 cases, the SARS-CoV-2 induces the elevation of pro-inflammatory cytokines and eventually forms a cytokine storm in the patient’s body, which probably damages the organs and cause death [12]. Clinical studies have proven that the SARS-CoV-2 infection is also capable of inducing cardiac injuries, kidney failure, acute renal, kidney, liver and cardiac injuries. It has been demonstrated that the average age of the incidence of severe COVID-19 associated with neutrophilia, leukocytosis and death is higher than the surviving patients, over 65 years old. By knowing these facts, identifying and considering these potential risk factors could help the physicians for a better prognosis and diagnosis [13,14].

From a diagnostic point of view, specific virus antigens or specific antibodies against these antigens are detectable in the specimens collected from COVID-19 positive patients e.g., respiratory swabs, saliva, blood, serum, stool and other types of samples [15]. Considering the guidance published by the World Health Organization (WHO), FDA and centers of disease control and prevention (CDC), nucleic acid (NA)-based tests are the main tests employed for detection of the virus in suspected cases and serological tests are mostly qualitative tools beneficial for confirmation of the reported result as supplementary tests together with other clinical data [16,17]. As the protein-based COVID-19 diagnostic strategies have demonstrated a broad range of accuracy, to measure the characteristics of each developed kit, i.e., sensitivity and specificity, their results are evaluated with a reference method. This reference panel is established by FDA and consists of standardized materials. The main technology for evaluation of the specific kits is Real-Time (RT)-PCR using nasopharyngeal samples. The related kits are distributed between the different companies, and they are requested to perform tests using both the reference RT-PCR kit and their developed device and report the results back to the FDA [18].

## 3. Serodiagnosis of SARS-COV-2

Serological tests have been serving as one of the main categories to detect infectious diseases over time by targeting the specific antibodies (Ab) or so-called immunoglobulins (Ig) secreted in the human body. These systemic immunoglobulin are secreted from white blood cells (WBCs) particularly B cells (lymphocytes) as protective proteins during the infection [19,20]. Each antibody contains four structural proteins Two heavy (H) chains and two light (L) chains when the N-terminuses of all of the chains are the antigen-binding site, specific to a particular antigen and different from the other antibodies. There are different isotypes of antibodies including immunoglobulin M (IgM), IgG and IgA which are distinguished by their specific regions in their heavy chains C-terminus. IgG as the most frequent antibody is secreted in the blood (serum), IgM is another antibody isotype in the blood, while IgA is abundant in both blood and other liquids in the human body such as saliva and breast milk. The expression of the specific antibodies binding to SARS-CoV-2 antigens is upregulated after the viral infection [21]. More specifically, IgM experiences an elevated expression in the moderate phase of the disease until about week two of the infection and begins to decline and almost disappears until week seven, while IgG expression is upregulated in the late phase from weeks 2–3 and remains high even beyond seven weeks (the exact durations are unknown) [22,23]. IgM detection has low sensitivity in the early stage of infection, which results in requiring repeated sample-taking every day. On the other hand, IgG is not preferred for screening the infection, but for patients’ follow-up, self-healing and convalescence status as well as the determination of the immune response of asymptomatic cases [24]. Compared with IgM and IgG, IgA has gained less attention for diagnostic purposes; however, the evidence suggests that IgA upregulation takes place earlier even before IgM and systematic studies on the IgA in COVID-19 patients are still lacking [25]. The first generations of the tests targeting SARS-CoV-2 antibodies applied SARS-CoV antigens to detect the SARS-CoV-2 specific Abs due to the absence of the specific antigens of the new coronavirus at that time. This problem was addressed after achieving more information regarding the new virus during the next months [26].

All of the serological assays rely on capturing specific antibodies in a mixture using an antibody–antigen attachment [27]. Based on the literature, the majority of the COVID-19 specific antibodies are against SARS-CoV-2 N antigen making them the most sensitive targets for serodiagnosis. The most specific antibodies are against the S1 domain of S protein. For this reason, S1 is suggested as the most specific viral target while S2 has demonstrated to have cross-reactions with SARS-CoV-1 specific antibodies. In the early days after infection, the sensitivity of serological tests is low and results in a false negative. The amounts of the antibodies are not high enough to be detected in the samples collected from the patients in the early phases of the infection, while, after about 10–15 days and in symptomatic cases, the tests demonstrate higher sensitivity and specificity to the desired antibodies [28]. It is noteworthy that the limitation of the serological tests that detect the specific antibodies is the probability of demonstrating cross-reaction due to the presence of pre-existing antibodies or for other reasons. Vaccination or prior infection with the SARS-CoV-2 could result in activating the immune system followed by secretion of long-term persistence of a portion of antibodies such as SARS-CoV-2 specific IgG [29]. The results of such tests strongly depend on affecting factors e.g., sample type, patient situation and disease phase at the time of the collecting sample. Generally, antibody tests are very informative and important owing to their ability for past infection detection, but they are not reliable for the early detection of COVID-19 in the first stages of the disease because seroconversion occurs after symptom appearance. These tests are more informative while performing for the evaluation of convalescence status, immune response and employed as screening tools for testing the rate of serosurvey and prevalence [30].

By the date, hundreds of serological tests have been developed and received FDA EUA approval to be performed for COVID-19 detection to target one specific Ab or a different combination of them in the blood or blood products of the suspected individuals either in a lab-based or in a POC setting [31]. Considering the fact that many of these technologies have been developed very recently, the provided information regarding the tests’ performance and accuracy are mostly based on both the companies and the FDA reports. By passing of time and the availability of more reports, we will cover the clinical reports of these tests along with their applicability in the real world in our future research.

### 3.1. Laboratory-Based Non-Isotopic Immunoassay (NIIA) Serological Tests

Enzyme-linked immunoassays (EIA) are NIIA assays that are suitable for lab-based detection or measurement of specific antibodies in the blood and serve as the main strategies which regularly assist laboratory scientists for COVID-19 screening. Generally, the main drawbacks limiting the clinical application of the immunoassays are high costs, their complexity and long duration making them not applicable as a simple and rapid POC and near-patient tool [32].

EIA variants including particularly Chemiluminescence Immunoassays (CLIA) and Enzyme-linked immunosorbent assay (ELISA) are frequently being employed for COVID-19 detection; they are also the routine serological methods for convalescence prediction and quantifying materials including antibodies and hormones. All of the immunoassays in this category take advantage of the affinity between antigens and specific antibodies with enzyme-based labels usually immobilized on the microplate surface [33].

#### 3.1.1. Chemiluminescence Immunoassays (CLIA)

CLIA is one of the most popular immunoassays which utilizes chemiluminescent or light-emitting labels for the detection of biomolecules present in blood, serum or plasma in a total duration of about 1–2 h [34]. In this technology, recombinant antigens labelled with chemiluminescent materials or luminescent substances form complexes with the specific antibodies when the positive sample is introduced to the microwell, followed by instrument-based detection for light-emitting signal measurement [35]. CLIA assays benefit from the advantages of automation, requiring a low amount of antigen and shorter sample-to-result time compared with some other serological strategies. This technology is known as a sensitive method to detect the small amounts of proteins with a high throughput; however, there might be some problems during the measurement or compound solubility. The main drawbacks of this strategy are being complex and costly compared with other serological tests [33]. The proposed CLIA-based diagnostic tests for SARS-CoV-2 detection have mainly used viral N or S antigens or a combination of them [36]. Based on the literature, the RBD domain of S and N antigen are the most preferred choices to be used for the detection of IgM/IgG or total antibody. The highest reported accuracy is related to the RBD-based CLIA detecting IgG antibody in the blood samples [37]. It should be highlighted that many factors such as the time of sample collection and disease phase affect the performance of these assays which might not be mentioned in the published articles. Although not being applicable in the POC setting, the main advantage of the fully automated versions of CLIA over rapid serological tests is the ability of high-throughput sample analysis [38].

Many researchers and manufacturers have developed diagnostic tests based on CLIA variants, electrochemiluminescence immunoassay (ECLIA) and chemiluminescent microparticle immunoassay (CMIA), to fight the current pandemic (Table 1). The main difference between ECLIA and CLIA is the chemiluminescence generation technique, electrochemical reactions in ECLIA and chemical reactions in CLIA. Infantino et al. evaluated the clinical accuracy of the Shenzhen YHLO Biotech CLIA kits for SARS-CoV-2 antibodies IgM and IgG. IgG demonstrated a lower cut-off for anti-SARS-CoV-2 antibodies [39]. Long and colleagues have studies IgM and IgG using a combination of S and N antigens in 363 samples from COVID-19 positive patients and observed a medium IgG- and IgM positive serostatus at day 13 after the symptom onset and a 100% seroconversion for IgG at day 20 [40]. Cai et al. proposed a Peptide-based Magnetic CLIA (MCLIA) for serological detection of COVID-19 and evaluated it with 276 sera samples from confirmed COVID-19 patients. The positive rate was 71.4% for IgG and 57.2% for IgM [41]. Lin and colleagues developed a chemiluminescence-immunoassay method using magnetic beads and recombinant nucleocapsid antigen for the detection of COVID-19. The test obtained a sensitivity of 60.76% for IgM and 92.25% for IgG. The specificity of the test for IgM and IgG was 92.25% and 97.5%, respectively. They concluded that IgG CLIA is more accurate compared with IgM CLIA and suitable to be performed with RT-PCR to improve clinical detection [42].

Another group employed the same assay with fewer specimens and reported low sensitivity of 48% but 100% specificity for the IgM test, while IgG detection illustrated 89% sensitivity and 91% specificity, respectively. By combining IgM and IgG testing, they achieved the highest accuracy for COVID-19 detection [43]. Ma et al. evaluated using RBD antigen and N antigen using CLIA and demonstrated a higher accuracy for the RBD-based CLIA detection. In addition, targeting IgA showed to be more sensitive and specific compared with IgM or IgG solely detection [37]. 

**Table 1 bioengineering-08-00054-t001:** Selected approved CLIA-based tests as Lab-based and high throughput COVID-19 detection strategies based on FDA Serology Test Performance [44].

Company	Test	Technology	Target	Antigen	Sensitivity (Day 15 after Symptom Onset)	Specificity	Throughput
Ortho Clinical Diagnostics, Inc.	VITROS Immunodiagnostic Products Anti-SARS-CoV-2 Total Reagent Pack	CLIA	Total Antibody	S	100%	100%	150 tests/h with one result in 48 min
Beckman Coulter, Inc.	Access SARS-CoV-2 IgG	Automated CLIA	IgG	S	96.8%	99.6%	50–200 tests/h
Babson Diagnostics, Inc.	Babson Diagnostics aC19G1	Fully Automated CLIA	IgG	S	100%	100%	440 tests/h
Ortho Clinical Diagnostics, Inc.	VITROS Immunodiagnostic Products Anti-SARS-CoV-2 IgG Reagent Pack	CLIA	IgG	S	90.0%	100%	150 tests/h
Siemens Healthcare Diagnostics Inc.	ADVIA Centaur SARS-CoV-2 Total (COV2T)	Automated-Semi-quantitative CMIA	Total Antibody	S	100%	99.8%	240 samples/h with one result in 18 min
Siemens Healthcare Diagnostics Inc.	Atellica IM SARS-CoV-2 Total (COV2T)	Automated CMIA	Total Antibody	S	100%	99.8%	440 tests/h with one result in 10 min
DiaSorin	LIAISON SARS-CoV-2 S1/S2 IgG	Complex Automated CLIA	IgG	S (S1/S2)	97.6%	99.3%	170 tests/h and 35 min time to first result
Vibrant America Clinical Labs	Vibrant COVID-19 Ab Assay	CLIA	IgM and IgG	S and N	98.1%	98.6%	24–36 h
SNIBE Diagnostic	MAGLUMI 2019-nCoV IgM/IgG	Automated CLIA	IgM and IgG	S and N	64.3%	100%	30 min for one test
Diazyme Laboratories, Inc.	Diazyme SARS-CoV-2 IgM CLIA test	CLIA	IgM	S and N	94.4%	98.3%	50 tests/h
Diazyme Laboratories, Inc.	Diazyme DZ-Lite SARS-CoV-2 IgG CLIA Kit	Automated CLIA	IgG	S and N	100%	97.4%	50 tests/h
Roche Diagnostics	Elecsys Anti-SARS-CoV-2	ECLIA	Total Antibody	N	100%	99.8%	300 tests/h
Abbott Laboratories Inc.	Architect SARS-CoV-2 IgG	CMIA	IgG	N	100%	99.6%	100 samples in 70 min
Abbott Laboratories Inc.	Alinity i SARS-CoV-2 IgG	CMIA	IgG	N	100%	99.0%	4000 tests in 24 h, with a 29 min time to first result

#### 3.1.2. Enzyme-Linked Immunosorbent Assay (ELISA)

ELISA is one of the most frequently used methods for serological detection of the infection with about 1–5 h. ELISA, as the manual version of the automated CLIA, is a microwell/plate-based assay employing immobilized capture antigen, and a secondary/tracer antigen which targets specific antibodies in the serum, plasma and/or whole blood samples [45], resulting in fluorescence or visible colour change in a chromogenic substrate by enzymatic activity qualitatively or quantitatively (Figure 1) [45,46]. The advantages of ELISA are being sensitive and specific, detecting both current and previous infection, lower costs compared with CLIA, and high-throughput, requiring simple facilities and determining selective isotype and antibody titers. ELISA has many capabilities such as being performed as a multiplexed or microarray-based test for parallel detection of various antibodies in a single sample [47]. On the other hand, some challenges such as sample preparation, requiring high sample volume, probability of false positives, antibody variability, manual procedures and high workload, probable cross-reactions along the duration of the tests have limited their applications, especially as rapid POC tests. Requiring trained personnel and delivery of the samples to the specialized labs make these tests even more time and labour-consuming and costly [33]. Although laboratories widely employ ELISA for coronavirus 2019 detection, only a few numbers of commercialized kits are based on this strategy which may be resulted from these limitations.

For ELISA-based COVID-19 detection, the researchers represent the immune response of the human body to the SARS-CoV-2 infection, indicating prior or recent infection. Based on the literature, IgM and IgG are recognized to be more upregulated compared to IgA and act as better targets in COVID-19 investigations. However, studies have demonstrated that IgA is also increased in response to SARS-CoV-2 infection and some tests developed to target IgA in blood or serum samples as well [48].

Although a few numbers of the assays employ full-length S antigen as capture molecules in the test, it is commonly referred to use specific shorter peptides from this protein such as RBD domain. A research group developed two different versions of ELISA assay for the detection of S-specific antibodies using full-length S protein and RBD domain. When evaluated the tests, the data revealed that the reactivity of both of the antigens was high, with a significantly higher reactivity for S antigen [49]. Okba et al. also evaluated the effectiveness of S antigen and its S1 and RBD domains using different ELISA kits. They observed that RBD and N proteins achieved the best results for the samples from mild patients and when they were tested using samples from the patient on day 14 after symptom onset, RBD ELISA had 100% sensitivity for IgG and 94% sensitivity for IgM, while, for N protein, the specificity of targeting IgG and IgM was 94% and 88% respectively. They also observed that, when the capture molecule is S1 and targets are IgA and IgG, IgA ELISA demonstrates a better sensitivity and IgG displayed a better specificity [50]. In another study, Zhang et al. simultaneously targeted IgG and IgM against SARS-CoV-2 and demonstrated that anti-S antibodies are more appropriate to be detected compared with anti-N antigens [51].

Companies such as Bio-Rad Laboratories, DRG Diagnostics GmbH, Euroimmun, IBL International and Epitope Diagnostics have developed manual ELISA tests for COVID-19 detection. Various FDA EUA approved, and CE marked ELISA kits, have been commercially available targeting IgM, IgG, IgA, IgG/IgM or total antibodies in the collected specimens, whose names and characteristics are listed in Table 2. Only one of these authorized kits has targeted total neutralizing antibodies against SARS-CoV-2 RBD in human serum and plasma using blocking ELISA [52]; cPass™ SARS-CoV-2 Neutralization Antibody Detection Kit is a 96-well format ELISA test which has an unknown sensitivity, and cross-reactions may occur for various causes such as the pre-existing antibodies [53].

Lassaunière et al. validated three ELISAs with serum samples and demonstrated the best performance from Wantai SARS-CoV-2 Total Antibody ELISA with 100% specificity and 90% sensitivity, Euroimmun IgA ELISA showed 93% specificity and 90 sensitivity and Euroimmun IgG ELISA displayed the lowest sensitivity of 65% with 96% specificity [54]. Beijing Wantai Total Ab ELISA is reported to achieve higher sensitivity (94.5% claimed by manufacture) compared with Beijing Wantai IgM and IgG ELISAs with 83% and 65% sensitivities, respectively [28], which was similar to the lower sensitivity of 65% for Euroimmun IgG ELISA as well. It can be concluded that total antibody ELISA tests display a better performance with higher accuracy and less cross-react compared with IgG ELISA, IgM ELISA or IgA ELISA [54].

Microsphere Immunoassay (MIA) is one of the ELISA variants using fluorescent material-labelled secondary antibody and magnetic carboxylated microspheres-virus antigen particle conjugates for detection of the antibodies in serum [55]. This technique implements both flow cytometry and ELISA and includes two main parts. MIA is newer and more accurate than ELISA; however, it requires expensive instruments and materials and is usually more time-consuming [56]. In one research, semi-quantitative MIA (FDA approved) was evaluated. An important observation was that, using this technique, SARS-CoV-2 antibodies were detected in 26.3% of the patients when they were at the hospital, but this number increased to 100% after 21 days from symptom initiation. Hence, this technique is reliable for the evaluation of immune response in convalescent and symptomatic patients but not for early detection of COVID-19 [57]. It is demonstrated that comparing MIA and ELISA are considerably more sensitive and specific than lateral flow immunoassays (LFIA) [55]. Conversely, Crook evaluated the sensitivity and specificity of two SARS-CoV-2 antibody tests using ELISA and LFIA and observed a higher sensitivity for LFIA devices compared with ELISA (65–85% and 55–70% respectively) and 93–100% specificity for LFIA versus 95–100% specificity for the ELISA test [58]. Rosenberg et al. tried to estimate SARS-CoV-2 cumulative incidence by conducting a developed and validated SARS-CoV-2 IgG MIA-based test. In summary, magnetic beads were coupled with viral N antigens. Labelled goat anti-human IgG secondary antibody was employed for microsphere-bound IgG antibodies detection using median fluorescence intensity (MFI). The test has 99.75% specificity and 87.9 sensitivity [59]. Fong et al. developed a microsphere-based antibody assay (MBA) for detection of anti-RBD and anti-S-specific IgGs and validated it using 294 serum samples. This test achieved 100% specificity for anti-NP IgG and 98.9% specificity for anti-RBD IgG. MBA seropositive rate for COVID-19 convalescence was 79.5% for anti-RBD IgG and 89.8% for anti-NP IgG with a shorter duration compared with EIA [60].

### 3.2. Lab-Based Fluorescence Immunoassays (FIA)

FIA is a quantitative fluorescent-labelled immunoassay, a biochemical technique that detects the attachment of the capture antibody and the desired analyte [61]. This technology employs a fluorescent material that emits energy or light as a fluorescent signal. In this technique, fluorescent dyes such as FITC provide the signal and a microplate fluorometer measures it [62]. Fluorescent microsphere immunoassay (FMIA) is used as a quantitative or semiquantitative method for COVID-19 diagnosis. Compared with ELISA, FMIA has the advantage of being more accurate and cost-effective, detects the infection using both serum and non-serum samples and there is no need for cycles of dilution for performing a semi-quantitative test [63]. This technology requires a small volume of sample and simultaneously detects multiple targets [64]. However, the number of developed tests using FMIA is lower than other techniques.

FMIA employs beads or microspheres (Luminex) coated with antigen, and the wells house hundreds of distinct sets of beads for the detection of unique antigens. Each microparticle is coloured with a unique combination of two various fluorescent dyes with various ratios as reporters. Additionally, two distinct lasers excite the beads during the test [65]. An FMIA-based test developed by Wadsworth Center, New York SARS-CoV Microsphere Immunoassay has successfully received FDA EUA authorization for antibody Detection. This test uses the full-length N protein as the target antigen for the detection of total antibodies. The sensitivity and specificity of the test are estimated 88.0% and 98.8%, respectively, for performing the test on day 25 after the onset of the disease. The important fact is that the sensitivity will be lower for the tests at earlier days of the infection [66]. Another FDA-approved FMIA test is developed by Luminex Corporation, namely xMAP SARS-CoV-2 Multi-Antigen IgG Assay. This assay targets IgG antibodies against three different SARS-CoV-2 antigens including N, S1 and RBD polypeptides. The throughput of the test is 96 and reports the results in 3 h using plasma or serum samples. PPA for this assay for serum samples while using MAGPIX^®^ (NxTAG^®^-enabled) system is 71.4% and 96.2% for the samples collected on days 8–14 and >14 from symptom onset, respectively, while the NPA is 100% [67].

GenBody, Inc. has developed a manual Colloidal Gold Nanoparticle-Based FIA-based immunoassay, GenBody FIA COVID-19 IgM/IgG, which detects IgG and IgM in the samples collected from the patients. The sensitivity of the test is 50% at Days 1–6, 91.7% at Day 7 and after that and its specificity is 97.5%. The duration of the test is 10–15 min and employing the reader is optional for monitoring the results [68,69]. iChroma COVID-19 Ab is another FIA developed using fluorescent-labelled conjugated. The assay demonstrated no cross-reaction with other respiratory pathogens and reported the results in 10–15 min with a sensitivity of 95.8–97.0% [70].

### 3.3. Rapid Serological Lateral Flow-Based Tests

Considering the critical and fatal effects of COVID-19, rapid and accurate diagnostic tests are critical not only for detection of the infection at the earliest stages but to monitor the disease and perform convalescence studies to control any outbreaks [71,72]. Compared with lab-based diagnostic strategies, rapid serological assays have gained much attention since they are applicable in the near-patient or even at home for prompt and simple wide screening and detection of the specific antibodies against the pathogens [73]. However, several intrinsic shortcomings are currently associated with them such as poor clinical accuracy, high dependence on the disease stage and lack of the ability to distinguish the neutralizing antibodies and reporting false results. Although the rapid diagnostic tests require an upgrade to overcome these limitations, they play a fundamental role in diagnostic and epidemiologic studies to be adapted as reliable tools to be employed in the future pandemic.

Almost all of the rapid serological tests for SARS-CoV-2 detection are cassette-based devices relying on LFIAs or so-called Immunochromatographic strip tests (IST). Lateral Flow Assays (LFAs) are paper-based platforms for POC detection of infectious diseases such as COVID-19 as portable, fast, user-friendly and easily operated tests without requiring complex instruments and technical training. Detection using LFAs is based on specific protein–protein interactions using a chromatographic system [31]. One of the advantages of this assay is the ability to implement multiple tests and control bands simultaneously to rapidly detect multiple analytes in a single sample [74]. Qualitative or semi-quantitative rapid serological in vitro diagnostics (IVDs) are significantly cost-effective and deliver the results within 5–30 min. Such devices do not require to be performed by trained laboratory staff and are applicable in hospitals, emergency rooms and other patient care settings [75].

In LFIA strips, the first step is using a colorimetric method to detect the presence of the specific analyte(s). For this aim, biorecognition elements are fixed in the test line(s), and a control line to confirm the validity of the test (Figure 2) [76,77]. LFIA coloured signal reporters or reader-based qualitative and semi-quantitative reporters including carbon material, fluorescent particles, Quantum Dots (QDs), enzyme, liposome, magnetic nanoparticles or other nanoparticles detect the presence of specific antibodies [76]. Qualitative LFIAs achieve more accurate results due to adapting the signal transducer. Various transducers like electrical, optical and magnetic readers are employed in these systems to produce digital signals by transforming the labels of the captured particle [78]. However, these quantitative tests are not able to detect multi targets simultaneously. Moreover, they are scanner-based and require expensive instruments for monitoring, which is not suitable for a POC test [79].

For POC detection of the infection, the potential monitoring strategy is relying on the colour change visible to the naked eyes. Hence, colloidal gold nanoparticle (AuNP) is usually the preferred reporter which is widely observed in the developed and commercialized rapid tests due to its long-term stability, easy operation, rapid onsite detection, low costs, no requirement for complex and expensive instruments and eye-reading results, being highly biocompatible and negligible biological toxicity. However, AuNP-based LFIAs are not capable of performing quantitative measurements. One other drawback of these strips is the lower sensitivity compared with reader-based reporters [80]. The overall structure and workflow of the LFIA rapid tests for COVID-19 detection are almost similar with some differences such as the type of the reporter, capture molecule(s) and selected target(s) have a crucial impact on the sensitivity and accuracy of the device [81]. 

Almost all of these tests require a small amount of 10–20 µL of the sample from suspected patients, and the results are reported in about less than 30 min (Table 3) [52]. Cellex qSARS-CoV-2 IgG/IgM Rapid Test, the first FDA EUA approved rapid serological cassette-based test, consists of a burgundy-coloured conjugate pad containing SARS-CoV-2 recombinant S and N proteins as antigens conjugated with colloidal gold (SARS-CoV-2 conjugates) and rabbit IgG-gold conjugates. NC membrane strip houses an IgG line (G Line) coated with anti-human IgG, an IgM line (M Line) coated with anti-human IgM and the control line (C Line) coated with goat anti-rabbit IgG [82]. MyBioSource is another company developing SARS-CoV-2 IgM/IgG Antibody Assay Kit targeting total antibodies against the N protein of SARS-CoV-2 [83]. One of the rapid serological tests with good performance is developed by BioMedomics targeting IgG/IgM antibodies with 10 µL of serum/plasma or 20 µL finger-pricked blood in as low as 10 min [84]. One other prospective assay is Pharmacy AG SARS-CoV-2 rapid providing the results in 20 min [85]. Panagiota I. Kontou and colleagues evaluated IgG and IgM tests based on ELISA, CLIA, FIA and LFIA in COVID-19 positive samples in a systematic review. The results illustrated that tests using S antigen are more sensitive than N antigen-based ones capturing antibodies. It was also demonstrated that IgG tests have better performance and show a better sensitivity when the patients are in the latest days after symptom initiation. However, the combination of both antibodies achieved the best result. ELISA- and CLIA-based tests achieved the best sensitivity of about 90–94%, this value ranged from 80% to 89% for LFIA and FIA [26].

**Table 3 bioengineering-08-00054-t003:** Selected rapid LFIA serological tests for COVID-19 detection and their performance.

Company	Assay	Target	Capture Protein	Technology	Sensitivity (Day 15 after Symptom Onset)	Specificity	Time to Result
ADVAITE, Inc. [86]	RapCov Rapid COVID-19 Test	IgG	Recombinant SARS-CoV-2 N antigen	Colloidal Gold Nanoparticle-Based LFIA	90%	95.2%	Visual Read/ 15 min
Beijing Wantai Biological Pharmacy Enterprise Co., Ltd. [87]	Wantai SARS-CoV-2 Ab Rapid Test	Total Antibody	Recombinant SARS-CoV-2 S antigen	Colloidal Gold Nanoparticle-Based LFIA	98.8%	100%	Visual Read/ 10–20 min
Salofa Oy [88]	Sienna-Clarity COVIBLOCK COVID-19 IgG/IgM Rapid Test Cassette	IgM/IgG	Recombinant SARS-CoV-2 S antigen	Colloidal Gold Nanoparticle-Based LFIA	93.3%	98.8%	Visual Read/ 10–20 min
Xiamen Biotime Biotechnology Co., Ltd. [89]	BIOTIME SARS-CoV-2 IgG/IgM Rapid Qualitative Test	IgM/IgG	Recombinant SARS-CoV-2 S antigen	Colloidal Gold Nanoparticle-Based LFIA	100%	96.2%—Cross-reactivity with HIV+	Visual Read/ 15 min
Healgen Scientific LLC [90]	COVID-19 IgG/IgM Rapid Test Cassette	IgM/IgG	Recombinant SARS-CoV-2 antigens (S, S1 subunit)	Colloidal Gold Nanoparticle-Based LFIA	100%	97.5%	Visual Read/ 10–15 min
Hangzhou Laihe Biotech Co. [91]	LYHER Novel Coronavirus (2019-nCoV) IgM/IgG Antibody Combo Test Kit	IgM/IgG	Recombinant SARS-CoV-2 antigens (S, S1 subunit)	Colloidal Gold Nanoparticle-Based LFIA	100%	98.8%	Visual Read/ 10 min
Hangzhou Biotest Biotech Co., Ltd. [92]	RightSign COVID-19 IgG/IgM Rapid Test Cassette	IgM/IgG	Recombinant SARS-CoV-2 antigen (S, RBD Domain)	Colloidal Gold Nanoparticle-Based LFIA	100%	100%	Visual Read/ 10–20 min
Megna Health, Inc. [93]	Rapid COVID-19 IgM/IgG Combo Test Kit	IgM/IgG	Recombinant SARS-CoV-2 N antigen	Colloidal Gold Nanoparticle-Based LFIA	100%	95.0%	Visual Read/ 10–20 min
Biohit Healthcare (Hefei) Co., Ltd. [94]	Biohit SARS-CoV-2 IgM/IgG Antibody Test Kit	IgM/IgG	Recombinant SARS-CoV-2 N antigen	Colloidal Gold Nanoparticle-Based LFIA	96.7%	95.0%	Visual Read/ 10–20 min
Assure Tech (Hangzhou) Co., Ltd. [95]	Assure COVID-19 IgG/IgM Rapid Test Device	IgM/IgG	Recombinant SARS-CoV-2 S1 and N antigens	Colloidal Gold Nanoparticle-Based LFIA	100%	98%	Visual Read/ 20 min
Cellex, Inc. [96]	Cellex qSARS-CoV-2 IgG/IgM Cassette Rapid Test	IgM/IgG	Recombinant SARS-CoV-2 S and N antigens	Colloidal Gold Nanoparticle-Based LFIA	93.8%	96.0%	Visual Read/ 15–20 min
TBG Biotechnology Corp. [82]	TBG SARS-CoV-2 IgG / IgM Rapid Test Kit	IgM/IgG	Recombinant SARS-CoV-2 S and N antigens	Colloidal Gold Nanoparticle-Based LFIA	99.8%	99.8%	Visual Read/ 15 min
Biocan Diagnostics Inc. [97]	Tell Me Fast Novel Coronavirus (COVID-19) IgG/IgM Antibody Test	IgM/IgG	Recombinant SARS-CoV-2 S and N antigens	Colloidal Gold Nanoparticle-Based LFIA	96.2%	99.4%	Visual Read/ 10 min

Although reporter nanoparticles in most of the rapid tests are gold, some researchers and companies have developed strips using different detection techniques. The DPP COVID-19 IgM/IgG test introduced by Chembio Diagnostics reporting the results in 15 min requiring optical readout using MicroReader 1 and 2 analyzers [98]. The FDA EUA had revoked the EUA of this test due to the effectiveness for IgM, the sensitivity of 50% and 93.3% for IgM and IgG, respectively [52]. Chen and colleagues are another group that developed a test using LFIA that uses lanthanide-doped polystyrene nanoparticles (LNPs) for anti-SARV-CoV-2 IgG detection in human serum [99].

Until now, rapid strips have been widely adapted for novel coronavirus detection; however, this strategy requires to be modified in the future to be more accurate and reliable. The investigations demonstrate that the accuracy of the LFIA tests is lower compared with qRT-PCR tests. As a critical shortcoming, many reports illustrate that many of the developed LF tests suffer from sensitivity lower than 70% [100,101,102,103]. FDA EUA given to some of the rapid LFIAs are even revoked due to low accuracy and false-negative results such as Autobio Diagnostics Co., Ltd.’s Anti-SARS-CoV-2 Rapid Test and Chembio Diagnostic Systems, Inc.’s DPP COVID-19 IgM/IgG System. Generally, due to their lower sensitivity and a higher rate of cross-reactivity with other respiratory pathogens resulting in false-negative and false-positive results, respectively, rapid LFIA tests are better to be conducted as supplementary tests to confirm RT-PCR results especially when the sample is reported negative [52]. In some cases, rapid LFIAs are comparable with the automated complex CLIA serological assays such as LIAISON SARS-CoV-2 S1/S2 IgG and compete with them due to their low costs and simplicity and similar accuracy.

### 3.4. Protein Microarray

Antibody microarrays, or so-called antigen microarrays, belong to the category of protein microarrays with the unique capabilities and taking advantage of a novel promising proteomic technology performing high throughput, multiplex and miniaturized tests to target low-abundant analytes in the samples [104]. ELISA and LFIA are capable of targeting single or a few proteins; conversely, protein microarrays provide a proteome-wide characterization of the present antibodies in response to SARS-CoV-2 antigens [105]. This strategy is preferred for profiling antibodies by enabling antibody screening using some or all of the proteins present in SARS-CoV-2 particles with a high resolution [106]. SARS-CoV-2 genome encodes 28 proteins including 5 structural, 15 nonstructural and 8 accessory proteins; specific polypeptides from these 28 proteins can be employed for the fabrication of SARS-CoV-2 specific arrays [6].

The main limitation of the protein microarrays is the higher turn-around time (TaT) than most of the serological tests and the total duration of this test takes less than 24 h including sample preparation to data analysis [107]. Since the initiation of the COVID-19 pandemic, researchers and manufacturers have developed lab-based microarrays to screen and capture the SARS-CoV-2 specific antibodies. Jiang and colleagues proposed an antibody microarray to profile the SARS-CoV-2 specific IgG/IgM convalescence responses. Firstly, the oligonucleotides related to all of the SARS-CoV-2 proteins including RBD of S1 subunit were obtained from GeneBank, synthesized and cloned in *E. coli* BL21. Then, a total of 18 proteins including proteins extracted from the sequences of N gene, S gene or other ORFs of viral RNA including E gene and nsp genes were spotted on the PATH substrate slide and formed a 2 × 7 subarray format. The test was evaluated using 29 serum samples collected from recovered patients, and the results clearly illustrated that S1 and N protein are suitable for detection, S1 with higher sensitivity than N protein. The responses of the antibodies to ORF9b and NSP5 proteins were also significant. The data provide insights in the field of vaccine development as well as diagnostics and therapeutics [108]. Wang et al. developed a SARS-CoV-2 proteome microarray by immobilized 15 amino acid-long peptides with 5-amino acid overlap to cover all the proteomes. The processing time is estimated 1.5 h for this array with an LoD of 94 pg/mL. This peptide-based SARS-COV-2 proteome microarray was capable of profiling antibodies and epitopes related to COVID-19 [106].

The developed and commercialized antibodies for CODIV-19 produced by this company and other manufactures along with their main characteristics are presented in Table 4. Quotient Limited company has announced a novel antibody array developed for COVID-19 detection. MosaiQ ™ COVID-19 Antibody Magazine FDA EUA approved a commercialized device to detect IgG and IgM antibodies against the Spike S1 protein secreted in response to SARS-CoV-2. MosaiQ platform allows disease screening of patient blood and produces a comprehensive result in about 35 min. The inputs for this array are anticoagulated blood samples centrifuged and loaded with the throughput of 3000 tests in 24 h. With the presence of SARS-CoV-2 specific IgG and IgM antibodies, they will bind to the appropriate probes and the positive results are visualized and interpreted by the instrument camera as black spots for COVID-19 samples. Each microarray generates a reaction on 132 printed biological markers. The required sample volume is as low as 5 μL for each test and, after reporting the first result, the other results will be available every 24 s [109,110]. The performance of this microarray for the detection of COVID-19 specific antibodies has recently been evaluated using serum samples from Blood Donation Screening Laboratory and demonstrated high clinical accuracy; before the day from symptom onset, the sensitivity of IgG detection was 71–80%, while this rate was increased to about 100% after day 15 which was superior to some other high-throughput available antibody assays such as EuroImmun (sensitivity: 71%), Abbott (overall sensitivity: 78%) or Roche (overall sensitivity: 76 %). The specificity of the MosaiQ^®^ test was also evaluated 100% and higher than the three other well-known tests [111,112]. This chemiluminescence-based kit is one of the most expensive and complex automated COVID-19 detecting technologies while offering the throughput of thousands of samples per day with a short duration of the test. The fact is that such complex and expensive technologies have demonstrated comparable accuracies with simple and low-cost tests which require no trained laboratory personnel. For the tests with comparable sensitivity and specificity, the rapid tests with a simple workflow and lower cost compete with the complex and costly tests requiring trained laboratory staff [113].

**Table 4 bioengineering-08-00054-t004:** The developed protein microarray-based tests for COVID-19 detection.

Manufacturer	Test	Target	Microarray Content	Sensitivity	Specificity	Format	Regulatory Status	Note
Quotient Limited SAÂ [29]	MosaiQ ™ COVID-19 Antibody Microarray	IgG, IgM directed to SARS-CoV-2 S protein	SARS-CoV-2 S protein antigens	Varies based on the phase of the disease (71–100%)	99.8%	High-throughput automated Immunoassay-Antibody employing enhancement reagent to enable silver to nucleate on the gold nanoparticles	FDA EUA—CE-IVD	35 min for the first microarray, 24 s for each next microarray.
PEPperPRINT GmbH [114]	PEPperCHIP^®^ SARS-CoV-2 Proteome Microarray	IgG, IgA, and IgM	The whole proteome of SARS-CoV-2 (GenBank ID: MN908947.3) translated into overlapping peptides	(No info)	(No info)	Manual-One single peptide array	CE-IVD	For vaccine development, or screen viral antigens to find and characterize immunodominant epitopes for in-vitro diagnostics research
PEPperPRINT GmbH [115]	PEPperCHIP^®^ SARS-CoV Antigen Microarray	SARS-CoV-2 specific Antibodies	S, N, M and E antigens	(No info)	no cross-reactivity	Manual- Containing three array copies per microarray, with 998 antigen specific peptides printed in duplicate	CE-IVD	including a two-day experimental workflow
PEPperPRINT GmbH [116]	PEPperCHIP^®^ Pan-Corona Spike Protein Microarray	Antibodies against S antigen	S proteins derived from seven coronaviruses translated into overlapping peptides	(No info)	(No info)	One array with 4564 peptides in duplicate	RUO	For Serum antibody fingerprint analysis, Immune monitoring and Epitope studies
Nirmidas Biotech, Inc. [117]	pGOLD™ COVID-19 IgG/IgM Assay Kit	IgG and IgM against S1 subunit and rbd domain of S	Three SARS-CoV-2 specific antigens	Sensitivity > 87% for IgM 5 days post symptom, ~100% for IgG and IgM 15 days post symptom onset	>99.5	Automated semi-Quantitative Microarray Based High Throughput ELISA-like COVID-19 array	RUO	48 samples with controls in each run, read by western blot reader or Nirmidas’ MidaScan™instument
Sengenics Corporation Pte Ltd. [118]	ImmuSAFE™ Respiratory Virus Protein Microarray	SARS-CoV-2 specific Antibodies	Multiple SARS-CoV-2 proteins, N from 5 other human Coronaviruses as well as Influenza A and B HA antigen subtypes	(No info)	(No info)	Manual or automated single and double-colour fluorescently-labelled antibody assay	RUO	The key application is for research and development purposes
Sengenics Corporation Pte Ltd. [119]	ImmuSAFE™ COVID+ Biochip Test	SARS-CoV-2 specific Antibodies	Multiple SARS-CoV-2 specific domains (N and S) including full-length and numerous truncated versions	(No info)	(No info)	Single-colour fluorescently-labelled antibody assay, and Dual-colour fluorescently-labelled antibody assays for quantitative analysis	RUO	24 arrays per slide (24 samples per slide)—Key applications are vaccine clinical trials and seroprevalence research studies.

Sinommune™ Antigen Multiplex Microarray is another serosurveillance array developed collaboratively by Sino Biological and Nanoimmune Inc. for COVID-19 detection. This microarray consists of nitrocellulose slides containing single array pads with hundreds of spots including pre-printed recombinant antigens which are absorbed onto the 3D nc slide. This array includes 65 viral antigens including S1, S2, S1 + S2, HE, N, S RBD and Plpro antigens specific for SARS-CoV-2 and other selected five groups of coronavirus family for investigating their reaction with SARS-CoV-2 specific IgG antibody [120]. ImmuSAFE™ is another company that has developed a patented technology to develop three different antibody microarrays using multiple domains, full-length and numerous truncated versions of SARS-CoV-2 S and N proteins. The chip determines different IgG, IgA, IgM antibodies and IgG1–4 subclasses. ImmuSAFE™ are also capable of assessing the response of the patient to the vaccines by differentiating the antibodies secreted in response to the vaccine, or they are a result of a previous infection [118].

## 4. Rapid Antigenic Tests

Antigenic assays are the second category of protein-based tests which are newly released to the market. This strategy relies on capturing the specific virus antigens in a mixture using an antibody–antigen attachment to detect the presence of the viral particles directly [121]. Currently, LFIA is the preferred technology for the development of rapid POC and at-home antigenic tools. The antigenic tests are the only tools that have been received to be used at home even without a prescription or requiring assistance from a specialist [122].

The clinical performance of these tools is hugely dependent on various factors and the patient’s situation. The best time window for sensing the viral particles is the first week after the infection. The viral load is elevated at this time and antigen tests demonstrate their best performance. This amount decreases during the time which results in dropping the accuracy of the antigen testing in the next stages of the disease [123]. Another factor is the sample type which could directly affect the test results. Most of the current antigen-detecting tests are based on nasopharyngeal specimens which is similar to the gold standard RT-PCR tests and the measurement of the accuracy is less complicated [124]. However, the rest of the kits test nasal samples with a few ones detecting the antigens in salivary specimens. The variation in the sample type increases the complexity of the evaluating procedure using the gold standard which recommends using nasopharyngeal swabs [125].

The most frequently present protein in SARS-CoV-2 structure is the N protein, an evolutionary conserved and highly immunogenic phosphoprotein. S protein, specifically in the S1 RBD subdomain, is another immunogenic protein on the viral particle surface with rare changes in the amino acids. All of the antigen detecting tests apply antibodies specific to SARS-CoV-2 proteins, more frequently N antigen, as capture molecules. Rapid antigen tests have some advantages over PCR such as lower costs and faster speed. They are highly specific for SARS-CoV-2 but demonstrate a low sensitivity; this is one of the main current limitations of the antigenic tests explaining why they may not detect all of the active coronaviruses [126].

As of 15 February 2021, the number of the developed antigenic tests have been far less than the serological tests (Table 5) [52]. A portion of the developed antigenic tests employ Colloidal Gold Nanoparticles (AuNPs) as the reporter for visual detection of the infection. Conversely, the rest of these kits require a specific instrument for the detection step. Although the visual detection is simpler and cost-effective requiring less equipment, the reader-based tests have the advantage of getting the results automatically from the analyzer and releasing the results sending messages and posting the results to the patient file by integrating Laboratory Information System (LIS) to their detecting system [127].

On May 8, FDA authorized the first antigenic test developed by Quidel Corporation, Sofia SARS Antigen FIA to perform tests in authorized laboratories and also POC settings. This test is a cassette-based LF immunofluorescent sandwich assay that detects viral N protein. Sofia2 or Sofia analyzer is required for qualitative detection. The results are reported in 15 min with 87.5% sensitivity. The test detects both SARS-CoV and SARS-CoV-2 but is not capable of differentiating them from each other [128]. Sofia 2 Flu + SARS Antigen FIA is an automated test reporting the results of influenza A, influenza B and COVID-19 in a POC setting in 15 min. This sandwich immunofluorescent test should be performed using Sofia 2 instrument and is capable of detecting N antigens from other pathogens in direct swab specimens. However, the test does not distinguish SARS-CoV-2 from SARS-CoV infected samples [129]. The third FDA EUA authorized antigenic test is a BD Veritor System for Rapid Detection of SARS-CoV-2 developed by Becton, Dickinson and Company (BD). This test employs chromatographic digital immunoassay and detects viral N protein in the samples taken from the patients in the first five days of symptom initiation. The assay monitoring is not visual and depends on the reader. This test is a rapid (approximately 15 min) chromatographic digital immunoassay for the direct and qualitative detection of SARS-CoV-2 antigens in nasal swabs. The sensitivity of this test is 84% and has a 100% specificity for COVID-19 detection [130,131].

Lin et al. have developed a POC microfluidic immunoassay for detection of IgG/IgM/ SARS-CoV-2 Antigen simultaneously in 15 min and evaluated its clinical performance using 28 healthy and 26 COVID-19 samples (Figure 3). They combined various biomarkers as targets to increase the accuracy of the assay. This sample-to-answer test requires 10 μL and 70 μL dilution buffer as indicated. While the sample is COVID-19 positive, SARS-CoV-2 biomarkers attach to the capture antibodies which are coated with FMS (fluorescent microsphere) and the formed complex is immobilized on the fluorescence test region via a second interaction due to antigen–antibody interaction. After a 10-min duration, the portable fluorescence analyzer reports the results. The achieved cut-off was 100 (T value) for antigen detection, and 200 for sensing each IgG and IgM antibody. It was also observed that serum samples have a significantly fluorescent value compared with a pharyngeal swab. The proposed assay was then tested with samples from patients in 1−7 days onset and over 14 days after symptom separately which demonstrated a growth in the T value while the time changed [142].

LumiraDx SARS-CoV-2 Ag Test takes advantage of microfluidic FIA, antibodies specific to SARS-CoV-2 N protein are applied in the FIA and target viral N antigen in the collected specimens. The test has demonstrated no cross-reaction with other respiratory pathogens such as SARS-CoV-2 and reports the results in 12 min as one of the fastest antigenic tests. However, like the BD Veritor System, the LumiraDx SARS-CoV-2 Ag Test does not present visual results and requires to be monitored using the LumiraDx Instrument. The sensitivity and specificity of the test are measured 97.6% and 96.6% with an LoD of 32 TCID50/mL [143]. BinaxNOW COVID-19 Ag Card developed by Abbott Diagnostics is an LFIA targeting virus N protein with high sensitivity of 97.1% and low LoD of 22.5 TCID50/mL. The obtained results can be uploaded in NAVICA, a smartphone application developed by Abbott company. By uploading the COVID-19 negative results to this application, NAVICA-enabled organizations such as banks or workplaces will easily have access to the COVID-19 status of the people [144,145].

PCL Inc. has also introduced the new PCL COVID19 Ag Rapid FIA test. This test has received a certificate from different regulators including CE-IVD. This device is a POC rapid and cassette-based fluorescent immunoassay targeting SARS-CoV-2 N protein as an antigen is oropharyngeal, nasopharyngeal and sputum samples. The results are achieved in as fast as 10 min and FLA Analyzer is used as a fluorescent reader. Based on the manufacturing, the LoD of the test is 1000 PFU (active viruses), and its sensitivity and specificity are 100% and 97.78%, respectively [146]. Another research group has targeted the SARS-CoV-2 protein via Field-Effect Transistor-Based Biosensor using a specific IgG antibody against the SARS-CoV-2 spike protein [147].

The Simoa SARS-CoV-2 N Protein Antigen Test has recently been given FDA EUA for COVID-19 detection. Among all of the approved antigen tests, this test is the only high throughput kit with a higher TaT of about 80 min, although results are reported in 150 min for 96 tests. This test is a Paramagnetic Microbead-based Immunoassay requiring an analyzer for interpretation but does not differentiate SARS-CoV-2 and SARS-CoV from each other in the infected specimens [148]. The Clip COVID Rapid Antigen Test is the other authorized tool that is developed in a smartphone-based setting for the interpretation and measurement of the luminescence signal emitted from the luminescent nanomaterials [131].

An important application of the rapid diagnostic tools is at home and near-patient testing. Fortunately, two rapid antigenic tests have recently been given FDA EUA to be applied as home tests with or without prescription, [52]. The Ellume COVID-19 Home Test respectively is the first FDA EUA authorized non-prescription fully at-home COVID-19 detecting test that can be completely performed at home with the patient to detect or follow up on the infection. This test employs fluorescent LFIA for the detection of N antigen of SARS-CoV-2 and requires a smartphone as the readout instrument to report the results [149]. The Clip COVID Rapid Antigen Test is another FDA EUA approved tool for at-home testing which requires a prescription. Interestingly, the PPA and NPA of the Ellume COVID-19 Home Test are obtained 91% and 96%, respectively, for the asymptomatic cases, and 96% and 100%, respectively, for the symptomatic individuals which are very promising [136]. With such at-home tests, the self-isolation of the infected people and the disease follow up take place in a limited area minimizing the viral spread and assistance of the medical care system which not only significantly reduces the rate of the infection in the community but also provides critical information regarding the immune system during the quarantine days.

## 5. Other Biosensors

Considering the increasing demand for rapid, cost-effective and accurate tests for COVID-19 detection, a wide range of technologies have been developed to overcome the shortages in the testing area [150]. Researchers have recently reported a variety of biosensing strategies such as electrochemical, optical, electrical, mechanical and piezoelectric biosensors for the detection of pathogens. Among various biosensing technologies, field effective transistors have attracted scientists’ attention due to the miniaturized size, fast and sensitive response and parallel sensing with the potential of being used in the POC setting [151]. Seo and colleagues have introduced a graphene-based FET-based biosensor for SARS-CoV-2 rapid detection [147]. A 2D graphene sheet has the advantage of high carrier mobility and electrode conductivity as well as large specific areas that make it a reliable material for sensing purposes [152]. In this COVID-19 FET sensor, the sensitive graphene layer on the device is coated with a commercially available IgG antibody against the SARS-CoV-2 S spike protein. The performance of IgG antibodies was first validated using ELISA and then they were used as a receptor for SARS-CoV-2 detection. The immobilization of the IgG antibodies was completed using a probe linker, 1-pyrenebutyric acid N-hydroxysuccinimide ester (PBASE), which is an efficient agent for interface coupling. The fabricated device observed a real-time response and successfully detected SARS-CoV-2 S in both cultured SARS-CoV-2 virus and transport medium used for nasopharyngeal swabs antigens with high sensitivity and LoD of 1 fg/mL with the ability to differentiate it from MERS-CoV [147]. Apart from the considerable sensitivity and low LoD, the measuring set-up requires a costly and low-throughput semiconductor analyzer; in addition, a high concentration of the antibody (250 µg/mL) is needed for the functionalization of the device.

COVID-19 has also been detected using field-deployable/portable plasmonic fibre-optic absorbance biosensors (P-FAB). This device is developed based on P-FAB with the LoD of detecting down to attomolar (10–18 M) protein concentrations. P-FAB technology monitors the changes in the intensity/absorbent or power loss in the light which is propagated in a multimode U-bent fibre-optic probe using a green LED and a photodetector. Two assays have been suggested for this aim, one using a labelled approach and another one is label-free. In the former, AuNPs are immobilized on the biosensor and then covalently conjugated with anti-N protein monoclonal antibodies (detector antibody) by thiol-PEG-NHS binding. Non-specific interactions are prevented by bovine serum albumin (BSA) treatment. These biofunctionalized probes will detect N protein of the SARS-CoV-2 in saliva samples in 15 min. On the other hand, the latter employs AuNP-labeled capture and detector antibodies in a sandwich immunoassay. The biosensor matrix is first coated with anti-N protein monoclonal antibodies on the U-bent fibre-optic probe and then treated with BSA. The sample should be mixed with anti-N protein antibodies-gold nanoparticles’ conjugates and introduced to the sensing area. The results will be achieved in 5 min. Among these two introduced technologies, the label-free assay is more promising due to its one-step response with no need for reagents, but its drawback is poor specificity. In general, the sensitivity and low LoD of the devices over LFAs are considerable, and P-FAB technology has the potential to be developed as a COVID-19 diagnostic test to fight the pandemic [153].

Bioelectric recognition assay has been used in the development of a novel rapid and portable cell-based biosensor for COVID-19 detection. This method is based on Molecular Identification through Membrane Engineering, in which human chimeric spike S1-RBD specific antibody, recombinant human IgG1, was inserted into mammalian Vero cells via electroinserting. By the presence of viral S proteins is positive samples, they attach to their specific antibody on the surface of the cells and make a unique difference in biorecognition elements’ electric properties. This hyperpolarization was measured and recorded by a cell-biosensor, a customized multichannel potentiometer with a polydimethylsiloxane (PDMS) layer containing eight holes on its electrode’s polyester part. After the application, for the readout section, the potentiometer was connected to a tablet and recorded the measurements. The test did not require any sample-preparation steps, and the LoD of the test was 1 fg/mL and showed no cross-reaction with SARS-CoV-2 N protein. The range of responses was semi-linear from 10 fg to 1 µg/mL [154].

PathSensors Inc. is another company developing its sensor, CANARY biosensor, for novel coronavirus detection using a cell-based technology in 5 min. The test is based on CANARY ™ technology. In summary, the test is made by genetically engineering B lymphocytes with bioluminescence from jellyfish and specific antibodies developed in mice. The engineered cells are designed to emit light when they are exposed and attached to a secondary pathogen [155]. When SARS-CoV-2 is present, it binds to the specific antibodies on the surface of the engineered cells, and the biosensors detect the viruses and report their presence by emitting light. The presence of the target pathogen is confirmed by measuring light output from the cell [156].

Convat project is one of the funded projects by the H2020 European Union Framework program with the main goal of developing a POC nanophotonic sensor based on silicon photonics interferometric technology and microfluidics lab-on-chip integration. For this aim, three distinct assays are in development, the first one for viral genomic analysis and two others for direct virus detection and serological testing. For direct detection of the virus, SARS-CoV-2 specific monoclonal antibodies and nanobodies are produced, linked to the chip for capturing the complete virus and evaluated using deactivated SARS-CoV-2 virus and real samples from COVID-19 positive patients. The test quantifies the viral load in the sample. For direct RNA detection, WHO recommended sequences for SARS-CoV-2 PCR detection were evaluated and three highly specific candidate sequences were selected as targets with a similarity of 100% to SARS-CoV-2 and 0% to other genomes (E, N1 and N gene) with no need for PCR amplification. Different complementary probes are immobilized on the chip to hybridize with the target viral-specific sequences. The device is evaluated with synthetic RNA targets, viral SARS-CoV-2 RNA and RNA of other coronaviruses and the target sequence in the E gene and N1 gene have an LoD of 1 nM and 3 nM, respectively. Direct virus detection takes place by immobilizing anti-S1 antibodies on the surface of the chip and evaluating of its affinity and specificity. The LoD is measured at 19 ng/mL for this test. For the serological test, the targets are viral N, S1 and RBD antigens, and the intact virus is detected. The device is evaluated using S serum samples and COVID-19 positive samples. The LoD value of this chip is equal to 446 FFU/mL. The tests are performed in 30 min [157].

Förster or fluorescence resonance energy transfer (FRET) is another method achieving high resolutions of detection in the range of 1–10 nm compared with optical techniques. Through FRET signals, protein–protein interactions, protein conformation changes and proteolytic cleaves can be studied even in living cells. For viral protein detection, FRET is capable of sensing protein–protein interactions such as antibody–antigen [158]. FRET technology relies on the energy transfer between a pair of donors and acceptor molecules in a distance-dependent manner. The interaction between two fluorescent-labelled proteins emitting specific colours while distancing, with an overlap in fluorescence emission spectrums, produces a novel third fluorescent colour that differs from the two initial emissions [159]. A useful technology for SARS-CoV-2 study and detection can be FRET-based biosensors. For this aim, viral proteins such as S can be fused to FRET pair-proteins. It also can be utilized to investigate the enzymatic reactions in human cells during COVID-19 infection [160]. SPR-based optical biosensors are other tools which have previously been developed for SARS-CoV detection and can be considered as the other potential tools to be developed for the accurate detection of the SARS-CoV-2 [161].

## 6. Discussion

In this review, we discussed the current and potential protein-based strategies which are developed for COVID-19 detection and have the potential to be adapted for the detection of other pathogens in future pandemics. Having many unsuspected asymptomatic COVID-19 carriers interacting with other people in the society increases the risk of infecting healthy people and makes the virus spread much more quickly. This situation may lead to problems such as overloaded clinics and hospitals. This fact has raised concern regarding not only new coronavirus but also to avoid such probable outbreaks in the future. Early detection and isolation of the positive cases is pivotal for controlling any outbreak. For this aim, the development of rapid and adaptable diagnostic tools plays a critical role to limit the spread of the virus in the earliest stages in the future to avoid such pandemics. Although vaccination has the potential of improving the immune state of our bodies, the challenge is that, after the initiation of each outbreak, at least a few months are required to develop a safe and effective vaccine and large-scale vaccination itself required a long time. More importantly, natural mutations in virus RNA may immunize the virus against the vaccine. For these reasons, mass production of sensitive and low-cost POC diagnostic tools is critical to saving thousands of lives during the first months of the upcoming outbreaks.

Although a large number of the COVID-19 tests have been developed over the last year, it is still difficult to recommend the advantage of each technology over the other ones due to the lack of sufficient clinical reports for most of these tests. However, the main goal of this review is to open the discussion and stimulate the questions for the Bioengineering community and also the designers who are interested in developing and contributing in the diagnostic technologies. For this reason, in each section, we have provided the performance, advantages and disadvantages of each strategy, and we hope this review opens new discussions in the development of the novel assays such as combining the current technologies with electronic biosensors or developing fully integrated devices. As an illustration, in the future, we expect to see the fully integrated lower-cost technologies adoptable for the detection of pathogens in the early stages of the spread to avoid any pandemics. Generally, the protein-based tests have been demonstrated to be very useful during the previous SARS and MERS pandemics, and, in the meantime, they are widely employed for COVID-19 diagnostics as attractive strategies especially for large-scale screening purposes. In conclusion, novel, reliable, accurate, prompt and adaptable protein-based strategies are urgently needed to assist us in the current pandemic and future hazards.

## Figures and Tables

**Figure 1 bioengineering-08-00054-f001:**
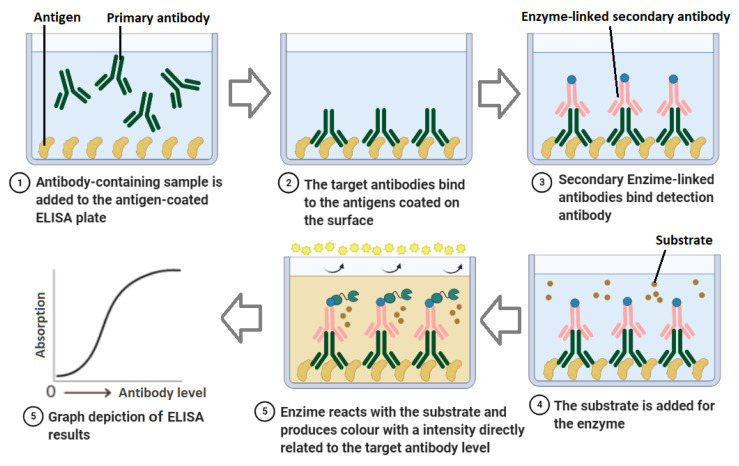
Schematic of ELISA technique for indirect detection of SARS-CoV-2. SARS-CoV-2 specific antibodies are added to the well and adhere to the immobilized viral antigens. Primary antibodies attach to the target antibodies, and a secondary enzyme-linked tracer antibody reacting with a chromogen is added and produces colour change. The intensity of the colour correlated with the concentration of the antigen in the sample.

**Figure 2 bioengineering-08-00054-f002:**
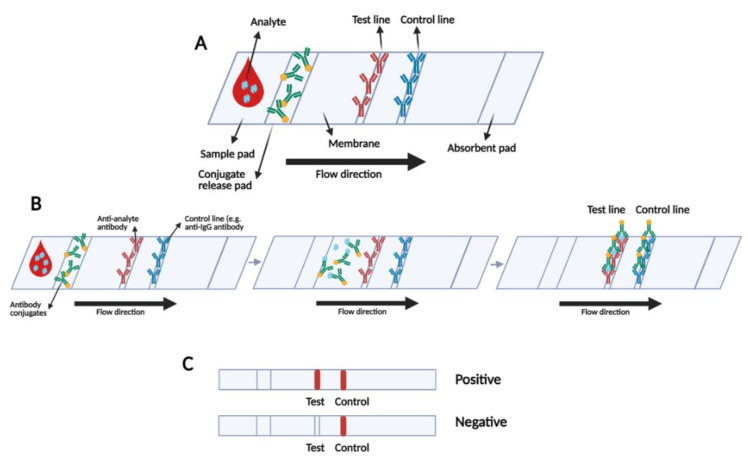
(**A**) Schematic of LFIA test strip; (**B**) mechanism of LFIA operation and (**C**) possible visual positive and negative results.

**Figure 3 bioengineering-08-00054-f003:**
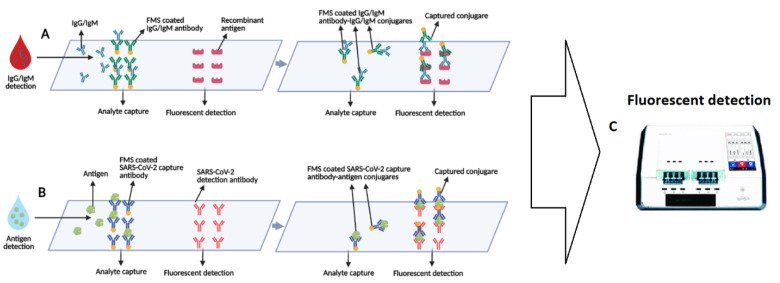
Schematic of the microfluidic fluorescence immunoassay for simultaneous detection of the antibodies and antigens. (**A**) Schematics of IgG/IgM detection and (**B**) schematics of antigen detection of SARS-CoV-2 using microfluidic FIA. (**C**) The results of both of the tests are simultaneously reported by using a portable fluorescence detecting device.

**Table 2 bioengineering-08-00054-t002:** Selected FDA EUA authorized Lab-based ELISA tests for COVID-19 detection [44].

Manufacture	Test	Target Antigen	Antibody	Technique	Sensitivity	Specificity
Mount Sinai Laboratory	Mt. Sinai Laboratory COVID-19 ELISA Antibody Test	Full length S antigen	IgG	High Throughput 2-Step direct ELISA	92.5%	100%
InBios International, Inc.	SCoV-2 Detect IgG ELISA	SARS-CoV-2 S antigen	IgG	High Throughput ELISA	100%	100%
InBios International, Inc.	SCoV-2 Detect IgM ELISA	S antigen	IgM	High Throughput ELISA	96.7%	98.8%
Siemens Healthcare Diagnostics	Dimension Vista SARS-CoV-2 Total antibody assay (COV2T)	S antigen	Total Antibody	Fully automated, fvfRapid High Throughput ELISA (10 min for one result, 440 assays per hour)	100%	99.8%
Quanterix Corporation	Simoa Semi-Quantitative SARS-CoV-2 IgG Antibody Test	S antigen	IgG	High Throughput, Automated Paramagnetic Microbead-based Sandwich ELISA. 96 tests in 2 h and 45 min	100% (LoD: 0.77 µg/mL)	99.2%
Beijing Wantai Biological Pharmacy Enterprise Co., Ltd.	WANTAI SARS-CoV-2 Ab ELISA	S antigen (RBD domain)	IgG	High Throughput ELISA	96.7%	97.5%
Emory Medical Laboratories	SARS-CoV-2 RBD IgG test	S antigen (RBD domain)	IgG	High Throughput ELISA	100%	96.4%
Siemens Healthcare Diagnostics	Dimension EXL SARS-CoV-2 Total antibody assay (CV2T)	S antigen (RBD domain)	Total Antibody	High Throughput ELISA	100%	99.9%
GenScript USA Inc.	cPass SARS-CoV-2 Neutralization Antibody Detection Kit	S antigen (RBD domain)	Total Neutralizing Antibodies	High Throughput Blocking ELISA (92 samples in 1 h)	100%	100%
EUROIMMUN US Inc.	Anti-SARS-CoV-2 ELISA	S antigen (S1 subunit)	IgG	High Throughput ELISA	90.0%	100%
Luminex Corporation	xMAP SARS-CoV-2 Multi-Antigen IgG Assay	S antigen (S1 subunit and RBD domain) and N antigen	IgG	Multiplex, microsphere-based and high-throughput FMIA (96 samples per run in each 3 h)	100%	99.2%
Bio-Rad Laboratories, Inc.	Platelia SARS-CoV-2 Total Ab assay	Recombinant N antigen	Total Antibody	High Throughput Semi-quantitative ELISA	92.2%	99.6%
Wadsworth Center, New York State Department of Health	New York SARS-CoV Microsphere Immunoassay for Antibody Detection	N antigen	Total Antibody	High Throughput MIA (FMIA)	88.0%	98.8%

**Table 5 bioengineering-08-00054-t005:** A list of the developed Antigen-based tests for COVID-19 detection.

Manufacturer Name	Test Name	Technology	Target Antigen	Sensitivity in Symptomatic Patients	Specificity	Detection	Test Duration
Abbott Diagnostics Scarborough, Inc. [132]	BinaxNOW COVID-19 Ag Card Home Test	Colloidal Gold Nanoparticle-Based LFIA, Prescription Home Testing	N antigen	97.1%	98.5%	Visual read + submitting the result via the NAVICA mobile application	15 min
Ellume Limited [133]	Ellume COVID-19 Home Test	Fluorescent LF, Over the Counter (OTC) Home Testing, Screening	N antigen	95%	97%	Instrument Read (smartphone-based)	15 min
Access Bio, Inc. [134]	CareStart COVID-19 Antigen test	Colloidal Gold Nanoparticle-Based LFIA	N antigen	88%	100%	Visual read	10 min
Princeton BioMeditech Corp [135]	Status COVID-19/Flu	Colloidal Gold Nanoparticle-Based LFIA, Multi-analyte	N antigen	93.9% (LoD: 2.7 × 103 TCID50/mL)	93.9%	Visual Read	15 min
Celltrion USA, Inc. [136]	COVID-19 Antigen MIA	Magnetic Force-assisted Electrochemical Sandwich Immunoassay (MESIA)	S antigen (RBD domain)	94.4% (LoD: 3.0 × 101 TCID50/mL)	100%	Instrument Read	10 min
Quanterix Corporation [137]	Simoa SARS-CoV-2 N Protein Antigen Test	High throughput Paramagnetic Microbead-based Immunoassay	N antigen	97.70% (LoD: 0.31 TCID50/mL)	Cross-reaction with SARS-CoV	Instrument Read	80 min
Luminostics, Inc. [138]	Clip COVID Rapid Antigen Test	LF immunoluminescent assay	N antigen	(LoD: 0.88 × 102 TCID50/mL)	Cross-reaction with SARS-CoV	Instrument Read (smartphone-based)	30 min
Abbott Diagnostics Scarborough, Inc. [139]	BinaxNOW COVID-19 Ag Card	Colloidal Gold Nanoparticle-Based LFIA	N antigen	97.1%/22.5 TCID50/mL	98.5%	Visual Read + submitting the result via the NAVICA mobile application	15 min
LumiraDx UK Ltd. [140]	LumiraDx SARS-CoV-2 Ag Test	FIA	N antigen	97.6% /32 TCID50/mL	96.6%	Instrument Read	12 min
Becton, Dickinson and Company (BD) [131]	BD Veritor System for Rapid Detection of SARS-CoV-2	Chromatographic digital immunoassay	N antigen	84%	No cross-reaction	Instrument Read	15 min
Quidel Corporation [128]	Sofia SARS Antigen FIA	FIA	N antigen	87.5%	Cross-reaction with SARS-CoV	Instrument Read	15 min
Quidel Corporation [141]	Sofia 2 Flu + SARS Antigen FIA	FIA	N antigen	(No info)	Cross-reaction with SARS-CoV	Instrument Read	15 min

## Data Availability

Not applicable.

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
