# Peer review of "COVID-19 Diagnostic Strategies Part II: Protein-Based Technologies"

_bioengineering, 2021, doi:10.3390/bioengineering8050054_

Round 1

Reviewer 1 Report

The presented paper is well organized and written about the protein-based COVID-19 diagnostic strategies in the commercial and medical field. Particularly, this paper properly covers the current technologies including lab-scale study and commecial products. After careful reading, I decide to accept this paper for publication in Bioenginerring.

Author Response

Dear Reviewer,

Thank you very much for your time and for reviewing this manuscript. We appreciate your positive feedback and accepting this paper in Bioengineering.

Sincerely,

Tina Shaffaf, Ebrahim Ghafar-Zadeh

Reviewer 2 Report

Summary and critique

This is an extensive review of the landscape of diagnostic tests that utilise protein-detection for SARS-CoV-2 infections. The authors have organised the review by category of analytes targeted for detection, ranging from host proteins (immunoglobulins) to viral proteins.  They have provided considerable detail of about each platform and overall, have covered the more promising assays in their tables and text.  The literature in this area of research has been adequately covered and major published papers are included in the bibliography. The technology underlying each platform is well-described.

However, the information provided about COVID-19 (the disease), the biology of SARS-CoV-2 and immune responses to the virus are basic and sometimes inaccurate (for example, referring repeatedly to peptides as ‘nucleotides’ in the section on protein microarrays). Clinical aspects of COVID-19 disease are covered very superficially. The results section, in addition to providing performance information, that has too-frequently been taken directly from the manufacturers, a review of this nature should provide ‘synthesis’ of the reported experience – for example, ranking of the assays by pre-defined characteristics.  All the published reports on performance of assays are treated as being similarly well-tested, although it is commonly agreed that the quality of reports vary widely in study design and statistical rigour.  The Discussion would benefit from recommendations by the authors for the best platforms and assays among the currently available platforms with their rationale for the choices.

Minor comments

  • References should be included within the tables where available (All tables)
  • When reporting sensitivity and specificity data for each assay, the ‘gold standard’ or reference test should be provided
  • The use of the term ‘nucleotides’ should be corrected when referring to peptides (Page 13)

Author Response

Dear Reviewer,

Thank you very much for your time and for reviewing this manuscript. You have mentioned very critical points which helped us to improve our manuscript. We tried to apply your valuable comments in the revised version in the best way.

Sincerely,

Tina Shaffaf, Ebrahim Ghafar-Zadeh

Reviewer 3 Report

The manuscript by Shaffaf and Ghafar-Zadeh is an extensive and interesting revision of protein-based SARS-CoV-2 assays and techniques. The review is easy to read, well-organized and deep, spanning from commercial, approved assays to newly developed and promising methods. There are few aspects that might be commented or clarified to improve understanding of the problems and advantages of the different techniques and assays:

- it might be helpful for laboratory researchers not aware with diagnostic evaluation methods to remind that sensitivity and specificity are calculated by comparison with a reference method, and the reference method in most of the works cited is RT-PCR in nasopharyngeal samples.

- it is important also to underline that antibodies appear several days or weeks after the onset of the infection, and the sensitivity of the serological methods will depend on the method but also on the patient population selected, the time of sample collection and the collection site (i.e. venopuncture or finger prick). So, the sensitivities of serological methods must be taken with care, and carefully reading the patient population and sample collection conditions.

- similarly, in the evaluation of antigen tests care should be taken concerning the samples used: some antigen assays are performed in nasopharyngeal samples (same as the reference method), but others are performed on nasal or saliva samples. In section 4 there is no mention about the types of samples used.

- it might be important also to remind that unless stated otherwise serological tests are not quantitative, even though they produce an index.

- in antigen tests that use readers it might be interesting to comment the advantages of sending the results through the LIS, while in those read visually the result must be introduced manually; this is important when performing hundreds of assays daily.

Minor points:

- revise spelling and English usage, although generally well written and easy to read, there are some errors:

- line 15: COVID-19 or SARS-CoV-2, not coronavirus 19

- line 81: insertion should read infection?

- lines 239, 245 "varies"

- lines 304-306, incomplete sentence?

- line 547 tested

- line 690 studied

- line 695 "results in the emotion of a varies third fluorescent colour", don't understand this

- line 705 asymptomatic

- line 713 improving the immune state?

- line 716 immunize?

Author Response

(The authors gave the same response as above.)

Reviewer 4 Report

The protein-based technologies have been demonstrated to be very powerful during the previous coronavirus pandemics, they are widely employed for COVID-19 diagnostics as attractive strategies specially for large-scale screening purposes.  In this review, the authors discussed the current and potential protein-based strategies which are developed for COVID-19 detection and have the potential to be adapted for the detection of other pathogens in future pandemics. The manuscript is well written, precise, could be interesting to the readers in “Bioengineering”.

Author Response

(The authors gave the same response as above.)

Reviewer 5 Report

Thank you for inviting me to peer review the very well written review titled "COVID-19 Diagnostic Strategies. Part II: Protein-Based Technologies" by Tina Shaffaf and Ebrahim Ghafar-Zadeh. The authors summarize available knowledge regarding protein-based diagnostic assays for COVID-19.

Comments:

  • Please clarify in the title, abstract and text that this is a narrative review and non-systematic.
  • The introduction is not very focused on the topic of the review - the detatils about COVID-19 vaccine are just confusing the reader.
  • On the other hand, it might be interesting to describe within the review how prior vaccination, prior infection, or different viral variants impact the results of the different protein-based diagnostic assays.
  • Finally, it would be very helpful to provide some information on the desing and methodological rigour of studies evaluating the diagnostic characteristics of each of the assays described.

Author Response

(The authors gave the same response as above.)

Round 2

Reviewer 2 Report

The authors have rebutted the critique of this reviewer primarily on the grounds that the review targets bioengineers and focuses on the technologies underlying  the currently approved diagnostic tests that detect relevant proteins.  Although they have responded to examples of errors that were in the comments, as stated in the review, these are just examples of other similar inaccuracies that need correction.  This reviewer would hope that the authors would not accept errors in one aspect of a subject just because they speak to a different expertise, just as engineer would refrain from accepting erroneous descriptions of technologies in discourse with biologists. Targeting a review to engineers does not remove the onus from the authors to convey  clear and accurate summaries of the biology of the system that is reviewed.

Unfortunately, careless errors remain throughout the revised manuscript which should be corrected with a careful proofreading of the draft,  examples include ‘SARD'-CoV-2 (lines 133 and 485).  Also noted is that errors are apparent in the illustration of some of the diagnostic technologies that is ostensibly intended for the engineering audience. For example, in Figure 3A the schematic suggests that fluorescence can be detected in the plate-bound antigens before the conjugated antibodies are captured.

Reviewer 5 Report

The authors have now addressed the peer review comments.